# MOBILITY NETWORKED TIME-SERIES FORECASTING BENCHMARK DATASETS

## ABSTRACT

Human mobility is crucial for urban planning (e.g., public transportation) and epidemic response strategies. However, existing research often neglects integrating comprehensive perspectives on spatial dynamics, temporal trends, and other contextual views due to the limitations of existing mobility datasets. To bridge this gap, we introduce **MOBINS** (**MOBI**lity **N**etworked time **S**eries), a novel dataset collection designed for networked time-series forecasting of dynamic human movements. **MOBINS** features diverse and explainable datasets that capture various mobility patterns across different transportation modes in four cities and two countries and cover both transportation and epidemic domains at the administrative area level. Our experiments with nine baseline methods reveal the significant impact of different model backbones on the proposed six datasets.

## 1 INTRODUCTION

Diverse and explainable human mobility datasets are crucial for advancing urban planning, affecting public transportation demand (Han et al., 2022), crowd congestion (Singh et al., 2020), traffic management (Liu et al., 2024), and infection prediction (Panagopoulos et al., 2021). Previous research focused on forecasting traffic and crowd congestion in specific areas using various transportation modes, such as subway systems (TianChi, 2019), ride-hailing services (Fivethirtyeight, 2015), and taxis (TLC, 2009). Additionally, there have been several attempts to predict COVID-19 infection by analyzing human mobility across different regions (Katragadda et al., 2022).

However, the datasets used in prior studies often fail to capture the diverse nature of human mobility from multiple perspectives. To comprehensively represent diverse mobility patterns, it is imperative to observe the movements of a large number of individuals over an extended period, taking into account various transportation modes. Unfortunately, many studies attempt to estimate demand using data either in a single transportation mode or in a short time frame (TianChi, 2019; Panagopoulos et al., 2021). Some efforts to understand human mobility rely on sparse movement data collected from a limited number of individuals. Despite the importance of understanding human mobility's impact on various aspects, such as transportation and epidemics, there is a lack of research that integrates additional information beyond transportation to enhance the diversity of mobility datasets.

Subway datasets (TianChi, 2019), a networked mobility dataset consisting of stations with high human traffic volumes, meet many of the specified criteria. Nevertheless, the subway datasets themselves do not offer multiple perspectives—i.e., diversity. Although there have been several studies to broaden a single data perspective (Shi et al., 2020), they only integrate mobility data from a *single* source with other contextual information that shares the same static topology. It is insufficient, for example, to simply add weather information as an additional variable to the time series. Instead, it is critical to use mobility-effected information at specific points of interest (PoIs) to *create synergy* between dynamic movements and networked time series. This approach not only enhances performance but also aids in understanding social phenomena that are difficult to discern from a single data source.

To improve the diversity of human mobility datasets, it is essential to collect data from different transportation modes across diverse regions over an extended period, capturing numerous daily movements. Moreover, incorporating additional contextual information, such as disease outbreaks, can aid in capturing various contextual patterns associated with spatio-temporal information. Meanwhile, for explainability purposes, the instances in the dataset should be organized on a network based on the spatial connectivity of each explainable area unit, such as an administrative area.

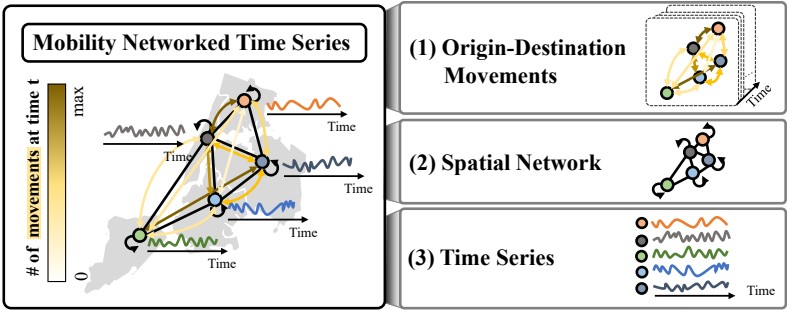

Figure 1: A structure of mobility networked time series in New York. **MOBINS** contains three components: (1) human movements from an origin to a destination over time, (2) spatial structure based on geographic proximity or a road network, and (3) time-varying features (e.g., numbers of taxi pick-ups and drop-offs) of each region. The first and third components cover the same period.

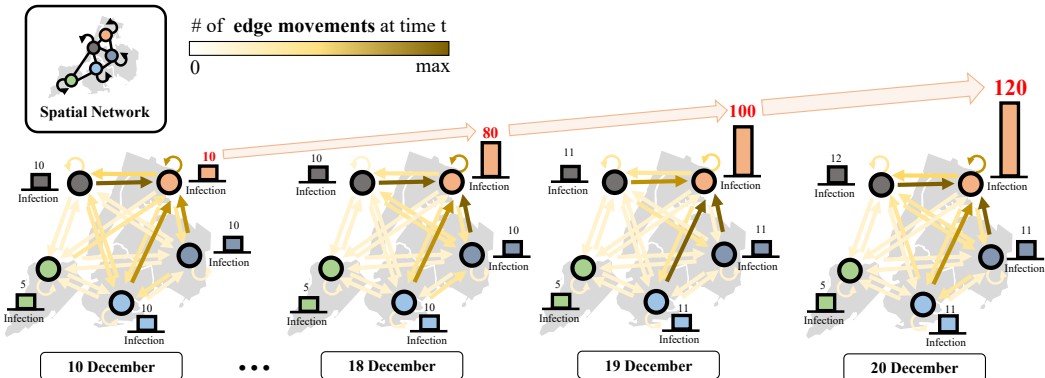

Figure 2: Dynamic edge movements and time-varying infection cases on a static spatial network. On top of the spatial network, node features represent the number of confirmed cases in each city or district over time, and edge features represent population movement flows between cities or districts over time. The increasing number of infection cases at the upper-right node is influenced by the increasing population flows to that node from other nodes.

Towards diverse and explainable human mobility datasets, we propose **MOBINS**, **MOBI**lity **N**etworked time-**S**eries forecasting benchmark. **MOBINS** offers a unique combination of origin-destination movements, a spatial network, and multiple time series, as illustrated in Figure 1. It involves multiple transportation modes including buses, subways, express buses, and taxis, providing a rich representation of human mobility patterns. With observations spanning at least two years and numerous daily movements, **MOBINS** enables the development and evaluation of advanced forecasting models. To ensure broad applicability, we include the benchmark datasets for transportation and infection prediction across four cities and two countries. By representing the networked mobility datasets at the administrative area level and treating each node as a distinct entity, **MOBINS** helps the model interpretation of the underlying mobility patterns.

Our dataset collection contains not only network-based interactions between nodes and edges but also temporal dynamics from time-varying features. Also, all datasets have a spatial network, where nodes represent locations such as stations, districts, and cities and edges represent connectivity between nodes based on subway lines, roads, and geographical adjacency. In Figure 2 visualizing part of a dataset in **MOBINS**, a spatial network created based on road network information is given as static data, and dynamic human mobility is represented through dynamic edge movements. In this case, the positive correlation between human movements and time-varying infection cases is captured. This kind of insight is difficult to uncover from a straightforward collection of multiple datasets, because their regions, spatial and temporal resolutions, and collection intervals may not be aligned. Therefore, this *new opportunity* clearly demonstrates the innovation and significance of **MOBINS**.

Our sophisticated and diverse dataset collection is publicly available together with forecasting methods. A substantial amount of time and effort has been dedicated to gathering comprehensive

datasets from various data sources, as well as merging and preprocessing them in preparation for their release. We aspire to contribute to the progress of the community that studies human mobility. Our contributions are as follows:

- **Datasets:** To the best of our knowledge, this is the first comprehensive dataset collection characterized by diversity and explainability for mobility networked time-series forecasting.
- **Experiments:** We conduct experiments to predict both time series and origin-destination movements. These experiments are based on various baselines with different backbones, applied to our dataset collection: transportation and epidemic datasets in four cities and two countries.
- **Takeaways:** Our experiments highlight the need for an integrated framework that simultaneously considers the three components—origin-destination movements, a spatial network, and multiple time series—in Figure 1. These insights guide future research directions in developing advanced frameworks for mobility networked time-series forecasting.

## 2 PRELIMINARIES

### 2.1 FORECASTING WITH TRANSPORTATION TIME-SERIES DATA

Human mobility prediction aims to predict each location's various attributes such as speed, demand, and congestion. In the context of traffic forecasting, studies employ traffic speed sensor datasets (Liu et al., 2024; Li et al., 2017) collected from PeMS (Performance Measurement System). Similarly, studies on demand or congestion prediction use modified inflow and outflow datasets derived from various transportation modes, such as subway (TianChi, 2019) or taxi (TLC, 2009) datasets. Unlike conventional time-series forecasting, mobility time-series forecasting emphasizes both temporal and spatial modules. Spatial axes are represented using $N \times N$ grids based on given coordinates, while an adjacency graph captures spatial connectivity derived from PoIs or a correlation generated from the sensor proximity (Jiang et al., 2021). Alternatively, station-based spatial connectivity is employed to model the patterns of movements within a given graph (Ou et al., 2020).

### 2.2 FORECASTING WITH ORIGIN-DESTINATION DATA

Origin-destination (OD) forecasting focuses on predicting the number of movements between the regions, capturing the interaction patterns within a mobility network. Datasets from ride-hailing services (Fivethirtyeight, 2015), taxi (TLC, 2009), and subway (TianChi, 2019) provide valuable information for deriving origins and destinations. OD movements between candidate origins and destinations, such as grids, stations, and PoIs, are forecasted using spatial and temporal modules (Han et al., 2022; Wang et al., 2019; Rong et al., 2023). Meanwhile, several studies have attempted to enhance time-series forecasting performance by incorporating OD movements. For example, research on COVID-19 prediction in England (Panagopoulos et al., 2021) and USA (Wang et al., 2023) has used the interaction between nodes, represented by the number of COVID-19 cases, and human mobility between regions. These studies leverage the relationship between inter-regional movement and the spread of infections to predict the number of cases in each region (Katragadda et al., 2022).

## 3 MOBILITY NETWORKED TIME SERIES

### 3.1 PROBLEM DEFINITIONS

Mobility is represented along both spatial and temporal dimensions. The spatial component is structured through a graph, denoted as $G = (V, E)$. The node set $V = \{v_1, v_2, \ldots, v_N\}$ captures locational data, while the edge set $E$ illustrates the connectivity between these nodes. Each node temporally aggregates *node time-series features* $X_t$, encompassing metrics such as transportation in/out-flow, ridership, infection rates, and additional time-sensitive data, where $X_t \in \mathbb{R}^{N \times d}$, $d$ is the number of feature variables, and $t$ is the index of the time. In scenarios where the graph $G$ remains static, its *spatial network* $A \in \mathbb{R}^{N \times N}$ is defined through a fixed adjacency matrix. Conversely, in dynamic settings, $G$ evolves with *OD movements* $M_t \in \mathbb{R}^{N \times N}$, where $M_t^{ij}$ accurately measures the volume of movements from node $v_i$ to node $v_j$ at each time point $t$.

Table 1: Comparisons based on the components of mobility networked time series (M: million).

| Datasets | | Spatial Nodes | Spatial Network Edges | Spatial Network Domain | OD Movements Daily Movements | OD Movements Modes | Node Time-Series Features Daily Amounts | Node Time-Series Features Domain | Time Period |
|---|---|---|---|---|---|---|---|---|---|
| Hangzhou Subway (TianChi, 2019) | | 81 | 85 | Station | 2.9M | Subway | 2.9M | Subway In/Out-flow | 01/01/2019 − 01/25/2019 |
| LargeST (CA) (Liu et al., 2024) | | 8600 | 201363 | Distance | - | - | 187.77M | Traffic Flow | 01/01/2017 − 12/31/2021 |
| COVID (England) (Panagopoulos et al., 2021) | | 129 | - | - | 11.86M | Mobile Device | 1975 | Infection | 03/01/2020 − 04/30/2020 |
| MOBINS (Transporation) | Seoul | 128 | 290 | Station-based Administrative Area | 2.68M | Smart Cards | 4.02M | Subway In/Out-flow | 01/01/2022 − 12/31/2023 |
| | Busan | 60 | 121 | | 0.63M | | 0.75M | | 01/01/2021 − 12/31/2023 |
| | Daegu | 61 | 123 | | 0.25M | | 0.34M | | 01/01/2021 − 12/31/2023 |
| | NYC | 5 | 12 | Borough | 0.10 M | Taxi | 3.03M | Ridership | 02/01/2022 − 03/31/2024 |
| MOBINS (Epidemic) | Korea | 16 | 45 | City & Province | 13.41M | Smart Cards | 25834 | Infection | 01/20/2020 − 08/31/2023 |
| | NYC | 5 | 12 | Borough | 2418 | Taxi | 2038 | Infection | 03/01/2020 − 12/31/2023 |

Table 2: Comparisons based on crucial criteria for mobility datasets.

| Datasets | Diversity (§3.2.1) | | | | | Explainability (§3.2.2) | |
|---|---|---|---|---|---|---|---|
| | Various Modes | Various Regions | Long Period | Many Daily Movements | Bi-Modal Dataset | Explainable Units | Spatial Network |
| Hangzhou Subway (TianChi, 2019) | X | X | X | O | O | O | O |
| LargeST (Liu et al., 2024) | O | O | O | - | X | X | O |
| COVID (Panagopoulos et al., 2021) | O | O | X | O | O | O | X |
| **MOBINS** | O | O | O | O | O | O | O |

**Definition 3.1** (MOBILITY NETWORKED TIME-SERIES FORECASTING). Given a spatial network $A$ and a corresponding historical dataset $D = \{D_1, D_2, \ldots D_T\}$, where $D_t = (X_t, M_t)$ includes node time-series features $X_t$ and OD movements $M_t$, the objective of *mobility networked time-series forecasting* is to learn a function $f$ that forecasts both the future node times-series features $\{X_{T+1}, X_{T+2}, \ldots, X_{T+H}\}$ and the future OD movements $\{M_{T+1}, M_{T+2}, \ldots, M_{T+H}\}$ over a forecast horizon $H$.

## 3.2 LIMITATIONS OF EXISTING MOBILITY DATASETS

Existing mobility datasets, as used in human mobility forecasting, are compared with the characteristics of our **MOBINS** in Table 1. We categorize existing human mobility datasets into three types. In the first type, the Hangzhou Subway dataset (TianChi, 2019) offers deep analysis through individual unit data but is limited by its specific region and short collection period, sharing the limitation also observed in datasets like the NYC Uber dataset (Fivethirtyeight, 2015). This dataset's collection from a single source makes it challenging to capture the diverse nature of human mobility. In the second type, LargeST (Liu et al., 2024) provides extensive data over a long collection period but lacks detailed human mobility information, such as OD movements. This limitation is also present in other PeMS-based datasets, such as the MERA-LA and PEMS-BAY datasets (Li et al., 2017). In the third type, Panagopoulos et al. (2021) shared human mobility datasets that link movements with other factors. However, its short collection period makes it challenging to observe long-term trends, and the absence of a spatial network reduces its utility for spatial analysis.

For urban planning purposes (e.g., public transportation) and epidemic response strategies, human mobility datasets should provide multiple views of spatial and temporal dimensions, as well as exhibit qualities such as diversity and explainability. However, many of the datasets currently available do not meet these criteria. In Table 2, we highlight the specific shortcomings of existing human mobility datasets, emphasizing their deficiencies in capturing essential qualities.

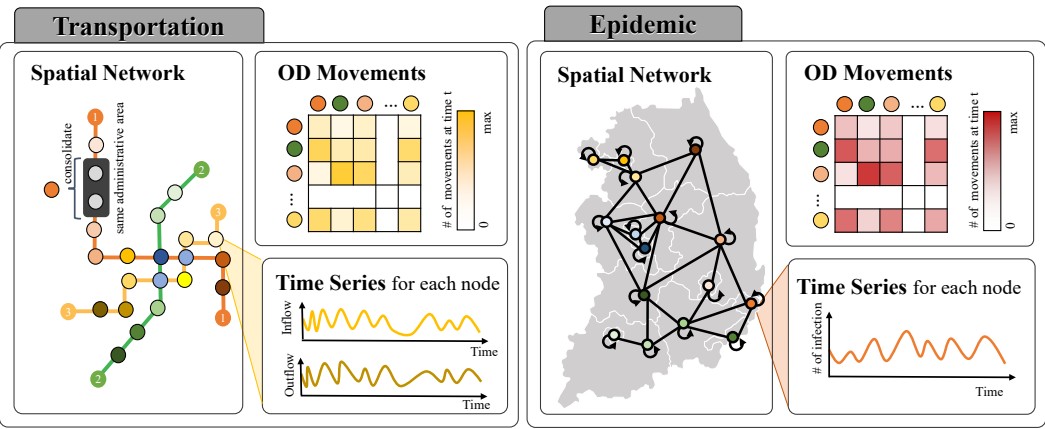

Figure 3: Composition of the transportation and epidemic datasets in South Korea.

### 3.2.1 DIVERSITY

To accurately represent human mobility, datasets should encompass a wide array of contexts. Human movement can occur through various modes of transportation, such as subways, city buses, long-distance buses, high-speed trains, taxis, personal vehicles, and ride-hailing services. A dataset that covers only a single mode of transportation, like the subway dataset (TianChi, 2019), fails to provide a comprehensive view of mobility. Datasets incorporating *various modes* are essential for depicting the diverse nature of human mobility. From a spatial perspective, the mobility datasets should encompass *various regions* to capture the different spatial and contextual patterns, such as commercial, residential, tourist, and mixed-use area patterns, emerging from diverse administrative areas. For instance, COVID datasets (Panagopoulos et al., 2021) cover four EU countries, and LargeST (Liu et al., 2024) includes datasets from across California, including Los Angeles, the Bay Area, and San Diego. From a temporal perspective, datasets should also include *long periods* to offer insights into both short-term and long-term mobility patterns. It is critical that datasets extend beyond simple metrics such as 'time of day' or 'day of the week' to include annual data, facilitating a richer temporal context. However, except for LargeST (Liu et al., 2024), many datasets cover periods of less than one year, with some training models over periods even shorter than one month (TianChi, 2019; Li et al., 2017). Moreover, mobility datasets must be collected with *many daily movements*. Unfortunately, several datasets are employed with only an insufficient number of daily movements (Wang et al., 2023), which fail to capture representative human mobility. Understanding human movements is not only about comprehending the movements themselves but also about linking information strongly correlated with these movements to get insights into social phenomena, which allows for the exploration of many aspects of human mobility. Therefore, *bi-modality* is helpful to comprehend human movements and their strongly correlated phenomena. For example, the COVID datasets consist of two types of data: OD movements from human mobility between regions based on mobile device data, and node time-series features from the number of infected individuals.

### 3.2.2 EXPLAINABILITY

Decision-makers in urban planning require models with high explainability, which necessitates datasets with inherent explainability. Training models using grid or sensor identifiers (Li et al., 2017; Liu et al., 2024) is insufficient. *Explainable units* for locational information, e.g., administrative areas, are vital. In the spatial dimension of mobility, each dataset should realistically represent spatial connectivity. For instance, the subway dataset (TianChi, 2019) records connectivity at the station level. Administrative areas can create a *spatial network* based on actual spatial adjacency and connectivity, indirectly helping to understand how the impact of an event spreads out.

## 4 DATASET COLLECTION: **MOBINS**

Our **MOBINS** dataset collection encompasses two domains: *transportation* and *epidemic*.

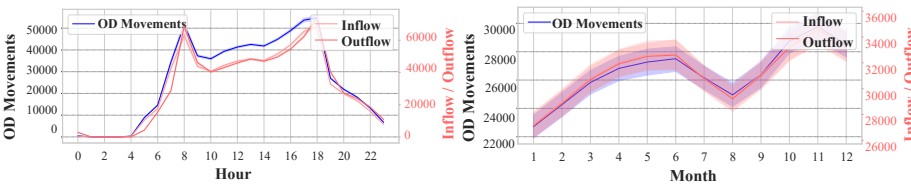

(a) [**Hours of the day**] Average and 95% C.I.  (b) [**Months of the year**] Average and 95% C.I.

Figure 4: Temporal patterns with positive correlations between inflow/outflow and OD movements about different periods in *Transportation-Busan*. Inflow/outflow and OD movements on all nodes are aggregated hourly or monthly to calculate the average and 95% confidence interval (C.I.).

## 4.1 DATASET CONSTRUCTION

**Transportation datasets:** The **MOBINS** dataset collection comprises transportation data from three South Korea cities (Seoul, Busan, and Daegu) and one U.S. city (New York City). The *Transportation-[Seoul, Busan, Daegu]* datasets include node time-series features from subway inflow/outflow data and OD movements from smart card usage across various public transportation modes. These datasets use subway maps to represent spatial connectivity, leveraging the commonalities between node time-series features and OD movements. However, pre-processing is required to align the data to a consistent spatial and hourly resolution, as node time-series features are generated for each station and OD movements are based on administrative areas. Figure 3 illustrates that stations within the same administrative area are consolidated into a single node in the spatial network, resulting in nodes represented by station-based administrative areas. The *Transportation-NYC* dataset includes OD movements from the NYC yellow and green taxi datasets (TLC, 2009) and node time-series features from NYC subway, tram, and railway ridership data. The spatial network is built at the borough level to alleviate sparsity from the huge number of nodes. Consequently, NYC taxi records from 263 zones and NYC ridership data from 428 stations are represented at a consistent resolution.

**Epidemic datasets:** The **MOBINS** dataset collection includes epidemic datasets that consist of node time-series features obtained from COVID-19 infection count and OD movements obtained from a smart card or taxi trip records in South Korea or New York City (NYC). The "Epidemic" section in Figure 3 illustrates the composition of the *Epidemic-Korea* dataset based on the spatial networks characterized by an adjacency matrix with diagonal ones representing the connectivity between cities and provinces. The OD movements from buses, urban rails, railways, and long-distance buses are used to represent inter-city or inter-provincial movements. However, islands are excluded due to their distinct transportation modes. Each node represents a city or a province, with COVID-19 infection cases recorded at each administrative area. Similarly, for the *Epidemic-NYC* dataset, node time-series features are based on daily infection cases from the five boroughs, while OD movements are comprehensively integrated from the NYC yellow and green taxi datasets (TLC, 2009).

## 4.2 DATASET STRAWMAN ANALYSIS

**Transportation datasets:** Figure 4 illustrates both the 'hours of the day' and 'months of the year' patterns in the *Transportation-Busan* dataset, using the long-term data collection spanning at least two years. The dataset exhibits a strong positive correlation between OD movements and node time-series features, as evident from the similar temporal distributions. Though these two modalities may show different values at a fine granularity, their aggregated trends coincide with each other, which confirms the validity of the dataset. Common temporal patterns include commuting patterns at 8 a.m. and 6 p.m., where both OD movements and inflow/outflow reach their peak values, as

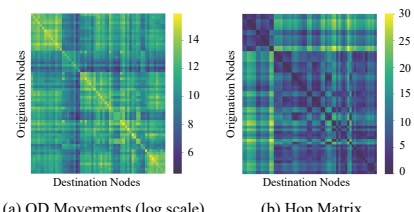

(a) OD Movements (log scale).  (b) Hop Matrix.

Figure 5: Spatial patterns of the OD movements and hop matrix in the *Transportation-Busan* dataset.

shown in Figure 4a. Also, these temporal patterns in Figures 4a and 4b highlight the importance of capturing both short-term and long-term dynamics in mobility networked time-series forecasting. From a spatial perspective, Figure 5a displays the total sum of OD movements between nodes, and Figure 5b is a matrix based on hops, indicating the number of nodes to be traversed from one node

(a) [**Korea**] Period: 01/01/2020 – 10/31/2021.  (b) [**NYC**] Period: 03/01/2020 – 03/31/2022.

Figure 6: Temporal patterns show negative relationships between infection cases and OD movements in the *Epidemic-[Korea, NYC]* datasets. The negative correlation is prominent in the yellow background. Infection cases and OD movements about all nodes are summed daily (M: million).

Table 3: Dataset statistics and default configurations. '# Node' is the number of nodes which indicate regions (e.g., stations or PoIs). We newly define forecasting target attributes with node time-series features and OD movements. For every node, the 'Target Dim.' is defined by $N^2 + d \cdot N$, where $N$ is the number of regions and $d$ is the number of feature variables from each node.

| Domain | Dataset | # Node | Target Dim. | Total Period | Train Days | Test Days | Time Interval |
|---|---|---|---|---|---|---|---|
| Transportation | Seoul | 128 | 16640 | 01/01/2022 – 12/31/2023 | 548 | 182 | 1 hour |
| | Busan | 60 | 3720 | 01/01/2021 – 12/31/2023 | 822 | 273 | 1 hour |
| | Daegu | 61 | 3843 | | | | |
| | NYC | 5 | 30 | 02/01/2022 – 03/31/2024 | 593 | 197 | 1 hour |
| Epidemic | Korea | 16 | 272 | 01/30/2020 – 08/31/2023 | 990 | 330 | 1 day |
| | NYC | 5 | 30 | 03/01/2020 – 12/31/2023 | 1051 | 350 | 1 day |

to another on the spatial network. Figure 5 reveals a negative correlation between OD movements and the hop matrix. In the hop matrix, darker colors represent a lower number of hops in the spatial network. Conversely, areas with higher (brighter) OD movements are associated with lower (darker) hops in the hop matrix. This finding suggests that a spatial network and OD movements are correlated, with higher mobility observed between nodes that have lower hops.

**Epidemic datasets:** Figure 6 presents the daily COVID-19 infection cases and daily OD movements for the *Epidemic-[Korea, NYC]* datasets. Figures 6a and 6b reveal a negative correlation between infection cases and movements during the early stages of the COVID-19 pandemic. As infection cases increase, human movements decrease, indicating a change in mobility patterns in response to the outbreak. From a temporal perspective, the *Epidemic-[Korea, NYC]* datasets demonstrate a strong negative correlation between node time-series features (infection cases) and OD movements, providing comprehensive insights into the interplay between the spread of infection and human mobility. This temporal analysis emphasizes the importance of considering the dynamic relationship between human mobility and disease spread.

## 5 EXPERIMENTS

Table 3 summarizes the statistics of the datasets used in our experiments.

### 5.1 EXPERIMENTAL SETTINGS

To evaluate our dataset collection with a four-day look-back window and various prediction lengths, we use Mean Absolute Error (MAE) as an evaluation metric, as shown in Table 4. We assess model performance across three different prediction lengths: 7, 14, and 30 days, to capture both short-term and long-term forecasting capabilities. Previous studies have employed prediction lengths ranging from 96 to 720 steps for long-term forecasting and 6 to 48 steps for short-term forecasting (Wu et al., 2022). For *Transportation-[Seoul, Busan, Daegu, NYC]* datasets that have a 1-hour time interval, we evaluate long-term forecasts at horizons of 168, 336, and 720 hours (i.e., 7, 14, and 30 days). Since the 1-hour interval results in many time points, these horizons are considered long-term. Meanwhile, for the *Epidemic-[Korea, NYC]* datasets, which have a 1-day time interval, the same prediction

periods of 7, 14, and 30 days represent short-term forecasts. Therefore, our dataset collection serves as a comprehensive benchmark for both long-term and short-term mobility networked time-series forecasting, depending on the datasets' time interval, with prediction lengths consistently set to 7, 14, and 30 days. For fair comparisons, all baselines are configured to follow the same experimental setup, running for 10 epochs with early stopping.

## 5.2 BASELINES

In our evaluation with **MOBINS**, we consider a broad range of traditional and modern forecasting models as baselines. We choose well-acknowledged prediction models as our benchmark, including (i) Linear-based models: DLinear, NLinear (Zeng et al., 2023); (ii) RNN-based model: SegRNN (Lin et al., 2023); (iii) Transformer-based models: Informer (Zhou et al., 2021), Reformer (Kitaev et al., 2020), PatchTST (Nie et al., 2022); (iv) CNN-based model: TimesNet (Wu et al., 2022); (v) GNN-based models: STGCN (Yu et al., 2018), MPNNLSTM (Panagopoulos et al., 2021).

Linear models take a linear approach to forecasting, treating time-series data as linear signals. DLinear and NLinear are known for their simplicity and efficiency, focusing on capturing linear trends and patterns. The RNN-based model, SegRNN, employs a recurrent neural network to capture temporal dependencies in the data. RNNs are well-suited for sequential data and are commonly used for time-series forecasting. SegRNN uses a segmentation-based technique to enhance the ability to capture long-range dependencies. Transformer-based models use self-attention mechanisms to capture long-range dependencies in time-series data. Reformer is an efficient variant of Transformer models that replaces dot-product attention with locality-sensitive hashing, reducing complexity and employing reversible residual layers to store activations only once during training. PatchTST extends the Transformer model to time-series data by breaking down the data into smaller patches. This approach allows the model to focus on localized patterns while leveraging the power of self-attention to understand broader trends. The CNN-based model, TimesNet, uses convolutional layers to capture temporal patterns in the data, allowing it to efficiently process time series with high-dimensional features. This model can identify localized patterns effectively, making it suitable for various time-series forecasting tasks. The GNN-based models use diverse graphs to deal with spatial information or integrate other contextual information for time-series forecasting. STGCN uses a static graph derived from inter-node proximity, while MPNNLSTM utilizes a dynamic graph based on OD movements.

## 5.3 BASELINE EVALUATION RESULTS

In this section, we outline the key results from our experiments, detailing how each baseline performs across a range of datasets. The outcomes highlight the relative strengths and weaknesses of different forecasting models and offer insights into their applicability in diverse contexts.

- **Linear models:** DLinear and NLinear demonstrated strong performance, achieving the lowest error rates on several datasets. DLinear was the best model for the *Transportation-Daegu* dataset across all prediction lengths and for the *Transportation-[Seoul, Busan]* datasets for 14-day and 30-day predictions. This result suggests that linear models can be highly effective in scenarios with simpler data patterns or lower degrees of complexity.
- **RNN-based models:** SegRNN showed competitive performance but did not achieve the best scores on any dataset, indicating that RNNs may face challenges with the increased complexity and longer-range dependencies typically associated with some time-series forecasting tasks.
- **Transformer-based models:** Recent approaches such as Informer, Reformer, and PatchTST were assessed. PatchTST excelled in the *Transportation-Seoul* dataset for the 7-day prediction length, achieving an average error rate of 0.3995 with a standard deviation of 0.0046. This result emphasizes the adaptability and versatility of Transformer-based approaches, which are known for their ability to handle long-range dependencies effectively.
- **CNN-based models:** TimesNet achieved the lowest error rates in several datasets, including the *Transportation-[Seoul, NYC]* and *Epidemic-[Korea, NYC]* datasets across all prediction lengths. These findings suggest that CNN-based models can be highly effective in certain contexts, particularly when dealing with spatio-temporal patterns.
- **GNN-based models:** STGCN and MPNNLSTM were evaluated, but they did not outperform other baseline models in any of the datasets. However, their performance was competitive, indicating

Table 4: Prediction comparison between nine baselines in terms of average **MAE** and standard deviation (in parentheses) with all prediction lengths (7, 14, and 30 days) in all datasets. The best model across each dataset is highlighted in **bold**. Please note the following abbreviations: "Pred." means "Prediction", "Trans." refers to "Transportation" and "Epic." denotes "Epidemic".

| Pred. day | Domain | Dataset | Linear-based | | RNN-based | Transformer-based | | | CNN-based | GNN-based | |
| --- | --- | --- | --- | --- | --- | --- | --- | --- | --- | --- | --- |
| | | | DLinear | NLinear | SegRNN | Informer | Reformer | PatchTST | TimesNet | STGCN | MPNNLSTM |
| 7 days | Trans. | Seoul | 0.3858 (±0.0068) | 0.4021 (±0.0003) | 0.7022 (±0.0363) | 0.9204 (±0.0018) | 0.5637 (±0.0315) | 0.3995 (±0.0046) | **0.3822** (**±0.0062**) | 0.4053 (±0.0047) | 0.6401 (±0.0009) |
| | | Busan | **0.5743** (**±0.0056**) | 0.5898 (±0.0006) | 0.9986 (±0.0087) | 3.4773 (±0.0031) | 0.7316 (±0.0075) | 0.6411 (±0.0052) | 0.6103 (±0.0642) | 0.6945 (±0.0032) | 0.9556 (±0.0035) |
| | | Daegu | **0.4677** (**±0.0004**) | 0.4919 (±0.0003) | 0.7876 (±0.0597) | 1.3885 (±0.0038) | 0.5338 (±0.0014) | 0.4916 (±0.0011) | 0.4902 (±0.0087) | 0.4901 (±0.0032) | 0.7337 (±0.0018) |
| | | NYC | 0.4491 (±0.0011) | 0.4460 (±0.0005) | 0.9226 (±0.0462) | 0.9147 (±0.007) | 0.5503 (±0.0036) | 0.4687 (±0.0027) | **0.3984** (**±0.0024**) | 0.4601 (±0.0019) | 0.6627 (±0.0015) |
| | Epic. | Korea | 0.5767 (±0.0031) | 0.5828 (±0.0015) | 0.5936 (±0.0072) | 1.7884 (±0.0013) | 0.7137 (±0.0392) | 0.6014 (±0.0392) | **0.4133** (**±0.0058**) | 0.7427 (±0.0199) | 0.7827 (±0.0062) |
| | | NYC | 0.4830 (±0.0016) | 0.4666 (±0.0022) | 0.4896 (±0.0179) | 1.0627 (±0.0015) | 0.5945 (±0.0165) | 0.5026 (±0.0044) | **0.3948** (**±0.0033**) | 0.5794 (±0.0038) | 0.6934 (±0.0062) |
| 14 days | Trans. | Seoul | **0.3878** (**±0.0047**) | 0.4072 (±0.0003) | 0.7183 (±0.0071) | 0.6453 (±0.0043) | 0.6310 (±0.0105) | 0.4006 (±0.0028) | 0.4015 (±0.0312) | 0.4182 (±0.0257) | 0.6399 (±0.0013) |
| | | Busan | **0.5830** (**±0.0075**) | 0.5934 (±0.0003) | 0.9913 (±0.0243) | 0.9482 (±0.0012) | 0.7434 (±0.0045) | 0.6324 (±0.0023) | 0.6175 (±0.0611) | 0.6862 (±0.0044) | 0.9528 (±0.0040) |
| | | Daegu | **0.4696** (**±0.0004**) | 0.4942 (±0.0004) | 0.8154 (±0.0039) | 0.7284 (±0.0004) | 0.5486 (±0.0045) | 0.4919 (±0.0007) | 0.4826 (±0.0033) | 0.4888 (±0.0021) | 0.7323 (±0.0009) |
| | | NYC | 0.4579 (±0.0023) | 0.4501 (±0.0004) | 0.9027 (±0.0237) | 0.7229 (±0.004) | 0.5623 (±0.0071) | 0.4680 (±0.0011) | **0.3988** (**±0.0017**) | 0.4629 (±0.0023) | 0.6624 (±0.0008) |
| | Epic. | Korea | 0.6258 (±0.0006) | 0.6088 (±0.0010) | 0.6484 (±0.0210) | 1.0182 (±0.0116) | 0.8025 (±0.0180) | 0.6467 (±0.0196) | **0.4562** (**±0.0063**) | 0.7726 (±0.0269) | 0.8003 (±0.0075) |
| | | NYC | 0.5008 (±0.0008) | 0.4784 (±0.0016) | 0.5341 (±0.0298) | 0.7046 (±0.0402) | 0.6012 (±0.0169) | 0.5100 (±0.0048) | **0.4026** (**±0.0033**) | 0.5855 (±0.0069) | 0.6970 (±0.0095) |
| 30 days | Trans. | Seoul | **0.3924** (**±0.0020**) | 0.5949 (±0.0001) | 0.7503 (±0.0708) | 0.6425 (±0.0006) | 0.6446 (±0.0059) | 0.4082 (±0.0034) | 0.4082 (±0.0095) | 0.4215 (±0.0075) | 0.6431 (±0.0016) |
| | | Busan | 0.5985 (±0.0023) | 0.6038 (±0.0004) | 0.9622 (±0.0453) | 0.9365 (±0.0024) | 0.7654 (±0.0241) | 0.6424 (±0.0028) | **0.5969** (**±0.0126**) | 0.6759 (±0.0015) | 0.9402 (±0.0001) |
| | | Daegu | **0.4750** (**±0.0004**) | 0.5006 (±0.0004) | 0.8132 (±0.0057) | 0.7285 (±0.0021) | 0.5849 (±0.0124) | 0.4957 (±0.0028) | 0.4846 (±0.0023) | 0.4923 (±0.0017) | 0.7315 (±0.0012) |
| | | NYC | 0.4747 (±0.0019) | 0.4592 (±0.0004) | 0.9075 (±0.0185) | 0.723 (±0.0013) | 0.5709 (±0.0122) | 0.4811 (±0.0022) | **0.4054** (**±0.0040**) | 0.4627 (±0.0045) | 0.6598 (±0.0005) |
| | Epic. | Korea | 0.7035 (±0.0028) | 0.6479 (±0.0012) | 0.7318 (±0.0504) | 1.0122 (±0.0077) | 1.1443 (±0.0469) | 0.7268 (±0.0197) | **0.5049** (**±0.0118**) | 0.8537 (±0.0500) | 0.8247 (±0.0172) |
| | | NYC | 0.5304 (±0.0014) | 0.4875 (±0.0010) | 0.5272 (±0.0286) | 0.7243 (±0.0138) | 0.6370 (±0.0121) | 0.5408 (±0.0068) | **0.4068** (**±0.0044**) | 0.6154 (±0.0189) | 0.6932 (±0.0104) |

that GNN-based approaches have the potential to manage complex network relationships and scenarios involving spatio-temporal interactions with more dataset-specific adaptations.

## 5.4 SUMMARY OF FINDINGS

Overall, the results indicate that no single forecasting model outperforms all others across all datasets. Instead, the choice of the best model depends on the specific characteristics of the dataset and the underlying data patterns. Linear-based models are effective in simpler scenarios, Transformer-based approaches excel in contexts with long-range dependencies, CNN-based methods work well with spatio-temporal data, and GNN-based models are ideal for datasets with complex networks.

- While linear models such as DLinear and NLinear perform well in simpler scenarios, they struggle with more complex data patterns and non-linear relationships. These models are limited in their ability to capture intricate temporal dependencies and are not suitable for datasets with highly dynamic or irregular patterns. However, in our datasets, they are simple but powerful baselines.
- RNN-based models, such as SegRNN, face challenges in handling long-range dependencies and complex temporal patterns. As the sequence length increases, RNNs suffer from vanishing or exploding gradients (Pascanu et al., 2013), limiting their effectiveness in capturing long-term dependencies. Therefore, SegRNN performs badly on our transportation datasets.
- While Transformer-based models like PatchTST demonstrate promising results in handling long-range dependencies, they struggle with capturing local patterns and short-term dynamics. The self-attention mechanism in Transformers can be computationally intensive, especially for longer sequences (Wang et al., 2020), which can limit their scalability. Moreover, Transformers often require large amounts of training data to achieve optimal performance.
- GNN-based models are designed to handle complex network relationships but require careful design and fine-tuning to achieve optimal performance. The performance of GNN-based models

heavily depends on the quality and representation of the graph structure, which can be challenging to construct for some datasets. Moreover, GNNs can be computationally expensive, especially for large-scale networks (Ding et al., 2022) and face scalability issues.

These findings provide a valuable reference for researchers and practitioners when selecting appropriate forecasting models for their specific applications. The comprehensive evaluation across diverse datasets and model architectures reinforces the importance of experimentation and context-driven decision-making in the field of mobility networked time-series forecasting. However, the limitations of existing forecasting models highlight the need for innovative approaches that can effectively address the challenges posed by complex and diverse datasets. That is, a novel approach is anticipated to outperform DLinear and TimesNet for this challenging problem.

## 6 FUTURE WORK AND LIMITATIONS

The complexity of mobility patterns requires diverse and comprehensive analysis for mobility networked time-series forecasting. Therefore, every component of mobility datasets captures spatio-temporal variability across multiple transportation modes and organizes the datasets into a bi-modal form, facilitating a comprehensive understanding of mobility trends over time. Additionally, the structure of the datasets with explainable units under a spatial network increases explainability, aiding decision-makers in interpreting mobility trends and implications for urban planning (Li et al., 2012; Hoang et al., 2016) and epidemic control (Ni & Weng, 2009; Katragadda et al., 2022) and these insights can significantly impact policy-making and economic decisions.

While **MOBINS** dataset collection serves as a forecasting benchmark, the presence of distribution shifts due to the changes in the *Epidemic-[Korea, NYC]* datasets suggests that they can be utilized for time-series online learning, adapting models in real-time. Additionally, the benchmark can be extended for research on imputation, clustering of traveling behaviors, and hierarchical time-series forecasting. Despite the advantages of our datasets, there are a few constraints, such as the fact that **MOBINS** is limited to only two domains and its period of dataset collection is mostly only two to three years, which is not enough to support annual patterns.

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
