# OpenReview forum: "Mobility Networked Time-Series Forecasting Benchmark Datasets"
_ICLR.cc/2025/Conference — ICLR 2025 Conference Withdrawn Submission_

### Official Review · Reviewer_QtiT · 2024-10-27

**Soundness:** 3
**Presentation:** 2
**Contribution:** 2
**Rating:** 5
**Confidence:** 4

**Summary:**

This paper proposes a mobility-networked dataset for the time series prediction task. In the dataset, the edge is defined by geographical proximity or human movement with different transportation modes.  And each node is associated with its time series. And benchmark results on  time series prediction are provided in the proposed dataset.

**Strengths:**

s1. Compared with previous data,  the released data also contains the movement information in the edge.
s2. The dataset contains a long time range of over 2 years.
s3. Benchmark results are conducted in the proposed dataset.

**Weaknesses:**

w1. One concern is about the space region of one node; considering an administrative region as a node in the graph is a little coarse. Especially,  for the NYC data, there are only 5 nodes.

w2. In terms of benchmarking, it would be better to include more State-of-the-Art GNN models for comparison.

w3. when introduce the dataset detail, it would be better to provide a table to show the detailed data fields.

w4. It would be better to add more introductions to point out what unique research opportunities brought by the ''human mobility'' modal.

**Questions:**

Please see the weakness above.

---

### Official Review · Reviewer_HA6j · 2024-11-01

**Soundness:** 2
**Presentation:** 3
**Contribution:** 2
**Rating:** 3
**Confidence:** 4

**Summary:**

This paper proposes a benchmark dataset MOBINS for mobility networked time-series forecasting. MOBINS offers a unique combination of origin-destination movements, a spatial network, and multiple time series. The benchmark dataset provides support for the interpretability of predicted results.

**Strengths:**

S1. This paper introduces a dataset collection MOBINS, which offers a unique combination of origin-destination movements, a spatial network, and multiple time series.
S2. Experiments show that the datasets can be applied in time series prediction and origin-destination movements.
S3. The dataset collection is available and is characterized by diversity and explainability for mobility networked time-series forecasting.

**Weaknesses:**

W1. This article organizes the existing data and integrates origin-destination movements, a spatial network, and multiple time series for prediction. However, the combination of OD and time series for training is not clearly stated. Is it possible to add comparative experiments using only a single data source and multiple sources of data?
W2. This article does not seem to make any other contribution besides organizing the dataset. The author does not explain the advantages of this dataset compared to other datasets in studying urban mobility.
W3. In Appendix Table 5, the line above NYC should be placed below.

**Questions:**

Q1. The research results show that Linear-based model and CNN-based model perform well. Does this mean that complex models are not suitable for this dataset?

---

### Official Review · Reviewer_E6zC · 2024-11-02

**Soundness:** 1
**Presentation:** 1
**Contribution:** 2
**Rating:** 3
**Confidence:** 4

**Summary:**

This work proposed mobility networked time-series forecasting benchmark datasets to facilitate mobility time series prediction. The paper makes significant contributions to transportation data analysis by introducing a dataset that spans multiple transportation modes and provides a long temporal range. However, its effectiveness is diminished by an overly complex presentation and a lack of detailed exploration into the unique aspects of the dataset. Furthermore, the paper would benefit from a more focused approach in its experimental design and the addition of a conclusion section to properly summarize its contributions and findings.

**Strengths:**

1. The paper provides a robust comparative analysis of various baselines, effectively demonstrating the proposed findings using their dataset.
2. It utilizes an extensive collection of time series data spanning at least two years, which enhances the reliability of the analysis.
3. The datasets include multiple modes of transportation, offering a comprehensive insight into transportation dynamics.

**Weaknesses:**

1. The paper introduces numerous concepts, which might overwhelm the reader. While the release of the South Korea smart card time series data is noteworthy, the integration with the readily available NYC Taxi data, and the attempt to simultaneously introduce a transportation and epidemic dataset, complicates the narrative unnecessarily.
2. The proposed dataset features fewer nodes compared to existing datasets, with only five nodes representing NYC. This limited division might not capture detailed, fine-grained information, potentially undermining the dataset’s utility.
3. The experimental section focuses solely on baseline comparisons, neglecting to clearly articulate the impact of different transportation modes—a claimed primary contribution of the dataset.
4. The omission of a conclusion section is a notable oversight, leaving the paper without a clear summary of its findings and implications.

**Questions:**

Please explain the weaknesses.

---

### Official Review · Reviewer_qsWE · 2024-11-03

**Soundness:** 2
**Presentation:** 2
**Contribution:** 3
**Rating:** 5
**Confidence:** 3

**Summary:**

In this study, a comprehensive mobility dataset is collected with benchmark evaluations over multiple deep learning models, e.g., RNN-based, Transformer-based, and GNN models. By comparing with existing dataset papers, the authors demonstrate the unique characteristics of this dataset, e.g., explainability. After evaluations of the proposed datasets, multiple findings are summarized to benefit future studies in this area.

**Strengths:**

[1] I appreciate the extensive experimental results that compare multiple types of prediction methods, which can benefit future studies.

[2] The authors summarize 4 findings that can provide guidance for future studies in this direction.

[3] The datasets collected by this study support downstream applications.

**Weaknesses:**

[1] One of my major concerns is the challenges behind this work. While authors have mentioned the diverse resolution between multi-source data is a challenge, it seems easy to address. Are there any other challenges?

[2] I am confused about the detailed composition of the dataset collected by this study. Does it mean for each city, there are multiple sources of mobility data available in the same period of time? e.g., taxi + bus + subway + epidemic. However I cannot read this info from Table 1.

[3] I don’t quite understand what messages Figure 1 is trying to tell us. Also in Figure 2, it seems the only difference between multiple subfigures is the increasing number from 10 to 120.

[4] Another concern is the intuition or motivation for collecting transportation + epidemic, rather than other data, given the diverse sources of mobility data, e.g., cellphone data from SafeGraph.

[5] In line 431 and 472, CNN is good at with spatio-temporal data, this is a vague statement.

[6] Could you explain more about the “explainability” brought by this dataset? This is highlighted multiple times in the paper with claims and statements.

**Questions:**

Please see comments above

---

### Note · Authors · 2024-11-13

I have read and agree with the venue's withdrawal policy on behalf of myself and my co-authors.